# A Systematic Review of Clinical Practice Guidelines on the Management of Malnutrition in Children with Congenital Heart Disease

**DOI:** 10.3390/nu16162778

**Published:** 2024-08-20

**Authors:** Maciej Kołodziej, Julia Skulimowska

**Affiliations:** Department of Pediatrics, The Medical University of Warsaw, 02-091 Warsaw, Poland; julia.skulimowska@gmail.com

**Keywords:** congenital heart disease, malnutrition, feeding guidelines

## Abstract

Congenital heart disease (CHD) is one of the most common inborn disorders, with a prevalence of 0.8–1.2%. Affected children are often malnourished due to increased dietary requirements. This may lead to severe long-term complications. Several authoritative organizations have published guidelines addressing nutritional intervention in children with CHD. We aimed to systematically assess the consistency of recommendations, the methodological quality of these guidelines, and the quality of evidence supporting each recommendation. PubMed, Embase, the Cochrane Database, World Health Organization Global Index Medicus, and 16 scientific societies’ websites were searched for the period until September 2023. The guideline quality was assessed using the AGREE II tool. After screening 765 records, only 2 guidelines published in 2013 and 2022 met our inclusion criteria. The main reason for exclusion was the absence of any system for rating the evidence. The main issues concerned the lack of implementation advice or tools and the lack of criteria to measure the application of guideline recommendations. The included guidelines were of good quality and within specific recommendations, both publications were largely in agreement, and the score for the overall assessment was high (83%). There is a pressing need for comprehensive, multi-threaded guidelines incorporating implementation strategies and methods for the performance assessment of children with malnutrition and CHD.

## 1. Introduction

Congenital heart disease (CHD) is one of the most common inborn disorders, with a worldwide prevalence of about 0.8–1.2% of all live births [1]. A significant group of children with CHD suffer from malnutrition, which is estimated to affect 30% of the pediatric CHD population [2]. The estimated prevalence of malnutrition also depends on the stage of interventional treatment of the disease, contributing to 27% in the preoperative period [3]. Improvement of nutritional status during the perioperative period is very important because it can directly improve the effects of the operation and rehabilitation and the length of hospitalization, reduce the number of complications, and lower the costs of care [4,5]. Cardiac surgery has a significant positive impact on patients’ weight gain and nutritional status [6,7]. The main reasons why children with CHD are susceptible to malnourishment are decreased dietary intake or increased metabolic expenditure, altered gastrointestinal perfusion, and the delayed acquisition of oromotor skills [8]. Malnutrition may lead to severe long-term complications, such as growth impairment, limitation of the child’s development, or increased mortality [9]. That is why it is important to follow nutritional protocols in this population, which may support children’s development and reduce the risks of malnutrition [10]. Despite the fact that a lot of clinical studies and narrative reviews are being conducted nowadays, there are very few universal, evidence-based guidelines that could be safely introduced in hospitals worldwide. Additionally, it seems that the least-explored topic is the nutrition of children in the preoperative period [11]. Considering the emerging evidence that proper nutritional preparation of the patient can significantly improve surgical outcomes and shorten the length of hospitalization (which translates to a reduced risk of perioperative and postoperative complications), it appears warranted for new research and guidelines to address the preoperative period in more detail [12]. The purpose of this paper is to systematically review existing guidelines on nutrition in children with CHD and to assess the consistency of their recommendations, the guidelines’ methodological quality, and the quality of evidence supporting each recommendation.

## 2. Materials and Methods

We followed the preferred reporting items for systematic reviews and meta-analyses (PRISMA) [13] statement and the systematic reviews of clinical practice guidelines laid down by Johnston et al. [14]. The protocol was published on the PROSPERO platform (CRD42023442138). For the quality assessment of the included guidelines, we followed the AGREE II user’s manual [15].

### 2.1. Data Sources and Searches

Using a strategy developed by an experienced librarian, two authors (J.S. and M.K.) independently searched PubMed, Embase, and the Cochrane Database of Systematic Reviews for all records published until 6 November 2023. Additionally, 16 scientific societies’ websites were searched. For the search strategy and a list of the searched websites, please see Appendix A. Due to the prolonged time needed for synthesis of the data, our search was updated in September 2023 and August 2024. No restriction, other than that they should be English-language publications, was used during this stage of the review.

### 2.2. Eligibility Criteria

We included in the search any clinical practice guidelines (CPG), or recommendations published in English, focusing on the management of malnutrition among children with CHD. According to the guidelines published by Johnston et al. [14], we developed our inclusion and exclusion criteria using the “PICAR” statement, which consists of five frameworks: population and clinical indications; conditions; interventions; comparators; and attributes of CPGs and recommendation characteristics.

Any consensus-based or expert opinion clinical practice guidelines without any system of rating evidence were excluded. Guidelines that were ongoing or unpublished were also excluded. Appendix A presents a detailed summary of the inclusion and exclusion criteria.

### 2.3. Data Extraction

Two reviewers (M.K. and J.S.) independently screened the titles and abstracts, full-text articles, and data extracted from the included guidelines by using standardized forms. Whilst abstracting the following characteristics were assessed: title, year, authors, language, organization, whether it was a novel publication or an update, the nutritional recommendations with the strength of each recommendation, and the authors’ assessment of the quality of supporting evidence. Both reviewers independently identified, extracted, and appraised the references to the evidence used to justify each recommendation, including the types of nutrition referenced in the supporting body of literature. Specific recommendations were summarized in a comparative table, focusing on possible gaps and common messages. The reviewers resolved disagreements by consensus.

The same two reviewers continued with an independent appraisal of guidelines by using the Appraisal of Guidelines for Research and Evaluation tool, 2nd edition (AGREE II).

### 2.4. Assessment of Guidelines Using AGREE II

An online tool, My AGREE PLUS (accessed on 6th September 2023 www.agreetrust.org) was used by two reviewers (J.S. and M.K.) to evaluate the guidelines. Both authors finished the online AGREE II tutorial before making the evaluation. The AGREE II instrument consists of 23 key items, which are grouped into 6 domains: (1) scope and purpose; (2) stakeholder involvement; (3) rigor of development; (4) clarity of presentation; (5) applicability; (6) editorial independence. It also contains 2 global rating items: (1) an overall guideline assessment (which is based on a general impression of a guideline after the completion of an evaluation of the 23 items) and (2) a question on whether the guideline could be recommended for use in practice (3-point scale: yes, yes with modification, or no). All the above-described key items and the overall guideline assessment item were assessed using a 7-point Likert agreement scale, which ranges from 1 (strongly disagree) to 7 (strongly agree). All scores that differed by 2 or more points between the reviewers were discussed until a consensus was reached. For each item and domain, the score was summed for both reviewers and then calculated as a percentage of the maximum possible score for that item and domain, using the formula provided by the AGREE II consortium:$$\frac{obtained\;score-minimum\;possible\;score}{maximum\;possible\;score-minimum\;possible\;score}\times 100\%$$

The possible scores ranged from 0% to 100%, corresponding to the minimum and maximum scores, respectively. The AGREE II instrument does not provide any range of scores to differentiate between high- and low-quality guidelines. For this reason, J.S. and M.K. agreed on a quality threshold of 60% for a domain score.

### 2.5. Statistical Analysis

Since only two guidelines were identified, no statistical analysis was conducted.

## 3. Results

A total of 765 records were identified in the search process, of which 602 were found in databases and registers, and 143 were found using other methods. After excluding the duplicates, 662 records were subject to further screening. Of these, 37 records were found to be eligible for full-text screening, leading to the final extraction of 2 guidelines that proved eligible for inclusion in this systematic review. For the flow diagram, please see Appendix A.

### 3.1. Characteristics of Included Guidelines

We included two guidelines published in 2013 and 2022 [16,17]. For their general characteristics, please see Table 1. Both papers were published by national organizations from the USA. Guidelines by Slicker et al. [16] focused only on patients with hypoplastic left heart syndrome (HLHS). In the study by Millis et al. [17], the authors focused on infants less than 6 months old with CHD who were admitted to an intensive care unit or inpatient ward. Millis et al. [17] and Slicker et al. [16] used different systems for rating evidence. Millis et al. [17] used the American College of Cardiology/American Heart Association (ACC/AHA) guidelines classification scheme [18], whereas Slicker et al. used a system adapted from Dellinger R.P. [19], wherein 1 means that the guideline is based on randomized control trials and 5 means that it is based on case series, uncontrolled studies, and expert opinion.

### 3.2. Quality of Included Guidelines (the AGREE II Quality Scores)

Table 2 provides individual as well as overall domain scores for the identified guidelines assessed using the AGREE II instrument. The scores varied, depending on the domain. Of all the domains, the highest mean was observed in the scope and purpose domain, as well as in the editorial independence domain, and was equal to 100%. The applicability domain presented the lowest domain score, with a mean of 21%.

### 3.3. Summary of Individual Domains

#### 3.3.1. Scope and Purpose (Domain 1)

In this domain, both guidelines achieved the highest score of 100%. In both guidelines, the overall objectives and health questions covered by the guidelines as well as the target population were specifically described.

#### 3.3.2. Stakeholder Involvement (Domain 2)

In the stakeholder domain, Mills et al. [17] received a slightly higher score of 37% compared to the 31% obtained by Slicker et al. [16]. The reason for such results was a lack of evidence on the views and preferences of the target population and a lack of a clear definition of the guideline target users, for which both Mills et al. [17] and Slicker et al. [16] received two out of seven points in all assessments. There was a one-point difference in the scoring of Item 4, i.e., the inclusion of individuals from all relevant professional groups in the process of guideline development, for which Mills et al. [17] received five and Slicker et al. [16] received four out of a maximum of seven points.

#### 3.3.3. Rigor of Development (Domain 3)

Both guidelines received comparable scores; Slicker et al. [16] received 67% and Mills et al. [17] received 72%. The main difference was in Item 9, which describes the strengths and limitations of the body of evidence, for which Mills et al. [17] received, on average, two points more than Slicker et al. Also for Item 8, which assesses the quality of the criteria for selecting evidence, Millis et al. [17] received the maximum score compared to Slicker et al. [16], who were given one point less. Otherwise, for Items 7 (use of systematic methods to search for evidence), 10 (description of methods for formulating the recommendations), 11 (consideration of the health benefits, side effects, and risks), and 12 (linkage between the recommendations and the supporting evidence), both Millis et al. [17] and Slicker et al. [16] received the maximum score of seven points. For both Items 13 and 14, both Millis et al. [17] and Slicker et al. [16] received one point, as the guidelines were not reviewed externally prior to their publication, and no information about updating the guidelines was included.

#### 3.3.4. Clarity of Presentation (Domain 4)

Both guidelines were noted to be high in terms of the clarity of presentation domain (94% and 100%). The difference was observed at only one point in the assessment of Item 15. This indicated the slightly higher specificity and ambiguity of the guidelines presented by Mills et al. [17].

#### 3.3.5. Applicability (Domain 5)

Mills et al. [17] received a slightly higher score for this domain of 25%, compared to the 17% obtained by Slicker et al. [16]. However, in both guidelines, this domain was the one that received the lowest score. This was due to the fact that none of the guidelines provided either advice and/or tools on how the recommendations could be put into practice (Item 19), on the potential resource implications of applying the recommendations (Item 20), or on the monitoring and/or auditing criteria (Item 21). The small difference in scoring lay in the assessment of Item 18, assessing the facilitators and barriers to the application of the guideline, for which Millis et al. [17] received 7/7 points and Slicker et al. [16] 5/7 points.

#### 3.3.6. Editorial Independence (Domain 6)

In this domain, both guidelines achieved the highest score of 100%. In both guidelines, the views of the funding body had no influence on the content of the guideline (Item 22) and there were no relevant conflicts of interest (Item 23).

#### 3.3.7. Overall Quality Score

Mills et al. [17] received a higher score for the overall quality of their guidelines (83%) compared to Slicker et al. [16], who received 67%. Slicker et al. [16] scored 111 points out of a maximum of 161 points, compared to Mills et al. [17], who received 118 points. For the specific scoring details, please see Table 2.

### 3.4. Summary of Recommendations

Table 3 provides a summary of the specific recommendations, listed separately for each recommendation or clinical indication. Not all recommendations were discussed in both publications. Below, we present a summary of the recommendations discussed in both publications, to assess their consistency.

#### 3.4.1. Resting Energy Expenditure/Starting Procedure

Both publications recommend indirect calorimetry as the first-line method for calculating REE. However, there was no agreement on the recommendations regarding using a stress factor in the process of calculating the REE. Enteral nutrition is recommended as the initial treatment option (if possible).

#### 3.4.2. Enteral Feeding

Enteral feeding is a safe and recommended route of feeding in hemodynamically stable patients; however, patients should be appropriately monitored during the process of feeding. Usage of nasogastric tubes or similar devices should be considered when oral feeding is insufficient or impossible. Human milk is recommended as a first-line choice for feeding.

#### 3.4.3. Calories

Recommendations regarding the goal for calorie intake levels during EN differed across the included guidelines: for Mills et al. [17], this was 90–120 kcal/kg/d, and for Slicker et al. [16], this was 120–150/kg/d. In this recommendation, the discussed guidelines differed quite significantly. This is likely due to the fact that the guidelines were published ten years apart (2013 vs. 2022) and because Slicker et al. addressed a specific patient population.

#### 3.4.4. Parenteral Nutrition

There was no consistency in the recommendations regarding the timing of parenteral nutrition (PN) introduction, Millis et al. [17] recommend PN when EN is insufficient and to delay its introduction if energy adequacy will be achieved during the first week of treatment, whereas Slicker et al. [16] recommend the early introduction of PN. Regarding calorie levels, the recommendations are also more consistent than those concerning enteral nutrition (>90–120 kcal/kg/day vs. 90–100 kcal/kg/day).

## 4. Discussion

This review aimed to systematically summarize the clinical guidelines for managing malnutrition in children with CHD. Only two guidelines were included. The small number of included studies, despite the many publications, narrative reviews, and recommendations on this topic, is caused by the fact that only two publications implemented a system for assessing the evidence to create the recommendation, and this was our criterion for inclusion in this review. We highlighted some significant differences and evolving recommendations over the past decade, as evidenced by Slicker et al. (2013) [16] and Mills et al. (2022), [17]. One of the reasons for significant differences between the publications can be attributed to the decade-long gap between their publication dates. Since 2013, there have been significant advancements in neonatal nutrition, medical technology, and our understanding of the metabolic needs of infants with CHD. Mills et al. [17] integrate these advancements, presenting guidelines that are informed by recent research and clinical experiences. The evolution in recommendations from Slicker et al. [16] to Mills et al. [17] highlights the dynamic nature of clinical practice and the importance of continuous guideline updates to incorporate the latest evidence.

### 4.1. Agreement with Other Publications

To date, many articles and reviews have been published that focus on the topic we are discussing here, both general recommendations and those for specific patient groups [10,20,21]. Despite the lack of a system for evaluating the evidence used, some of them formulate specific dietary recommendations, and some of them are consistent with those included in this review, but there are also some differences.

#### 4.1.1. Nutritional Interventions and Growth Outcomes

In the publications by Herridge et al. [22] and Maynord et al. [10], the authors emphasize the significance of early nutritional interventions to improve growth and overall outcomes in infants with CHD. Both studies support the use of enteral feeding as a primary strategy and underscore the necessity of aggressive nutritional management. Herridge et al. [22] specifically highlight the role of tailored nutritional support to address the high metabolic demands and feeding challenges faced by these infants. Similarly, Maynord et al. [10] demonstrate improved weight gain and reduced reliance on gastrostomy tubes with their structured nutritional program. This is consistent with the findings of Millis et al. [17], who also advocate for enteral feeding but recommend adjustments based on individual patient needs.

#### 4.1.2. Implementation of Nutritional Pathways

Marino et al. [20] developed a consensus-based nutritional pathway using a modified Delphi process, which aligns with the findings of Herridge et al. [22] and Maynord et al. [10] by stressing the importance of structured nutritional protocols. Marino et al.’s [20] pathway, which was designed to optimize preoperative and postoperative nutritional care, reflects a growing recognition of the need for systematic approaches to managing malnutrition in CHD patients. The consensus on early and structured nutritional intervention found in these studies supports the general recommendations of Millis et al. [17], although the specific implementation details differ. Since this is one of the most important components of the guidelines, any new documents should particularly focus on creating an appropriate strategy for implementing the recommendations.

### 4.2. Disagreement with Other Publications

#### 4.2.1. Variability in Caloric Goals

One notable disagreement is on the recommended caloric intake levels. Millis et al. [17] suggest a range of 90–120 kcal/kg/day, while Slicker et al. [16] propose a higher range of 120–150 kcal/kg/day. This discrepancy reflects a broader debate in the field about optimal caloric goals for this population. Herridge et al. [22] and Maynord et al. [10] also highlight the variability in caloric needs among CHD patients, suggesting that individual adjustments may be necessary based on patient-specific factors. The updated guidelines of Millis et al. [17] may incorporate more recent evidence on caloric needs compared to the older guidelines of Slicker et al. [16].

#### 4.2.2. Timing and Use of Parenteral Nutrition

Millis et al. [17] and Slicker et al. [16] present different approaches regarding the introduction of parenteral nutrition (PN). Millis et al. [17] recommend delaying PN if enteral nutrition is anticipated to meet energy needs within the first week, while Slicker et al. [16] advocate for the early introduction of PN. This difference reflects an ongoing debate about the optimal timing for PN in managing malnutrition in CHD patients. Herridge et al. [22] also discuss the role of PN, suggesting that its use should be individualized based on the patient’s response and needs.

#### 4.2.3. Implementation and Applicability

The guidelines by Millis et al. [17] and Slicker et al. [16] both lack detailed implementation strategies and tools. Marino et al. [20] address this gap by providing a structured nutritional pathway and implementation guidelines, which approach contrasts with the more general recommendations of Millis et al. [17] and Slicker et al. [16]. The absence of detailed implementation advice in the older guidelines points to the need for more actionable and practical frameworks, as emphasized by Marino et al. [20].

### 4.3. Strengths and Limitations

For this study, we followed rigorous guidelines by means of the PRISMA statement [13] and used the AGREE II tool for quality assessment [15]. This structured methodology ensures a comprehensive evaluation of the guidelines, enhancing the reliability of the findings. The use of multiple databases and scientific society websites strengthens the search process, reducing the risk of missing relevant guidelines. We identified only two guidelines that met the inclusion criteria, limiting the analysis’s scope. This small number may not fully represent the scope of current practices and recommendations in the field. The exclusion of guidelines without a system for rating evidence further narrows the scope, potentially omitting relevant guidelines that could offer additional insights. Although the AGREE II assessment indicates high-quality guidelines, there may still be potential biases in the development process of the included guidelines. For instance, the guidelines could be influenced by the preferences and practices of the organizations involved or the specific patient populations on which they focus. Further research into the development processes and potential biases in guideline formulation could provide additional insights. Given the rapid advancements in the medical and nutritional sciences, there is a need for continuous updating of guidelines to reflect the latest evidence and practices. The review highlights that while the included guidelines are of high quality, their recommendations may evolve as new research becomes available. Ensuring that guidelines remain current and relevant is an ongoing challenge that needs to be addressed to maintain their effectiveness in clinical practice.

## 5. Conclusions

The review of guidelines by Slicker et al. (2013) [16] and Mills et al. (2022) [17] reveals significant advancements in the nutritional management of infants with CHD over the past decade. While both sets of guidelines emphasize the importance of adequate caloric intake and prefer enteral nutrition, Mills et al. [17] offer a more comprehensive and updated framework that reflects current best practices. Both the evolution of treatment of children with CHD and malnutrition and the low number of guidelines on this topic create a pressing need for comprehensive, multi-threaded guidelines, incorporating implementation strategies and methods for the performance assessment of children with malnutrition and CHD.

## Figures and Tables

**Table 1 nutrients-16-02778-t001:** General characteristics of the identified guidelines (including sections contained in the PICAR framework).

Author, Year	Mills et al., 2022 [17]	Slicker et al., 2013 [16]
Working group or organization/country	Neonatal Cardiac Care Collaborative (NeoC3)/USA	Feeding Work Group of the National Pediatric Cardiology Quality Improvement Collaborative/USA
Methods used to determine recommendations	Literature review (American College of Cardiology/American Heart Association Clinical Practice Guideline Recommendation Classification System methodology) and consensus	Consensus through literature review and survey of participating centers
Population	Critically ill full-term neonates (≥37 weeks’ estimated gestational age and ≤ 28 days old) and infants (>28 days old) up to 6 months of age with CHD (structural, myopathic, or arrhythmic) admitted to the ICU or inpatient ward	Patients with HLHS from birth through the first interstage period
Classification of target population	Dependent on the cardiac lesion, hemodynamic support, and clinical parameters	Dependent on hemodynamic stability
Areas of recommendation	(1) Energy needs, (2) nutrient requirements, (3) enteral nutrition, (4) feeding practice, (5) parenteral nutrition, and (6) outcomes	(1) Preoperative enteral feeding, (2) parenteral nutrition, (3) postoperative enteral feeding, and (4) interstage feeding
Comparator	None	None
Financial support	The Neonatal Heart Society (Abbott Formula, Mead Johnson, Cheisi, Mallinckrodt, Prolacta, and Medtronic	None
Conflict of interest	None	None

**Table 2 nutrients-16-02778-t002:** Domains, scores, and overall scores of identified guidelines using the AGREE II tool.

Guideline Author, Year	Agree II Scaled Domain Scores						Overall Score
	1—Scope and Purpose	2—Stakeholder Involvement	3—Rigor of Development	4—Clarity of Presentation	5—Applicability	6—Editorial Independence	
Mills, et al., 2022 [17]	100%	37%	72%	100%	25%	100%	83%
Slicker et al., 2013 [16]	100%	31%	67%	94%	17%	100%	67%
Mean	100%	34%	70%	97%	21%	100%	75%

**Table 3 nutrients-16-02778-t003:** Comparison between the main recommendations contained in the identified guidelines.

Area of Recommendation	Mills et al., 2022 [17]	LOE And Class	Slicker et al., 2013 [16]	LOE and Grading (1–5)
Resting Energy Expenditure (REE)/Starting procedure				
REE	Determination is through IC, using the Schofield equation without the stress factor. If IC is not feasible, using estimating equations to approximate REE is an acceptable alternative. Equations without the stress factor are recommended.	IIa/C-LD IIB/C-EO	Determination is through IC with the stress factor (55 kcal/kg/day × 1.2–1.4)	5
Starting procedure(s)	Preoperative EN (conditions), when insufficient + other conditions → PN	-	TPN + IL → EN when there is hemodynamical stability and no contraindications	-
Enteral Nutrition (EN)				
Preoperative safety	EN may be reasonable when considering the type of cardiac lesion, hemodynamic support, and clinical parameters. EN may be considered if the patient is hemodynamically stable and in a monitored setting. EN may be considered in patients that are stable or on decreasing hemodynamic pharmacologic support.	IIb/C-LD IIb/C-EO Ib/C-EO	Suitable for safe and hemodynamically stable patients, although appropriate monitoring is required.	Recommended/2
Use of umbilical arterial catheters	EN feeding is possible and should not prevent the initiation of EN	IIb/C-LD	EN feeding is possible and should not prevent the initiation of EN	Strongly recommended/4
With PGE infusion	NS	-	EN feeding is possible	Strongly recommended/3
In ductal-dependent CHD	If the patient is hemodynamically stable and in a monitored setting	IIb/C-LD	EN feeding is possible	Strongly recommended/3
Fortified foods	May be beneficial	IIb/C-EO	NS	-
On ECMO/VAD	Recommended in hemodynamically stable and well-supported patients on mechanical circulatory support	IIb/C-EO	NS	-
Gastric/postpyloric feeds	In safe patients (postpyloric feeds may shorten time to the calorie goal)	IIb/C-LD	Nasogastric tube feeding may be utilized to deliver enteral feeding and can be used postoperatively to initiate EN	No recommendation/3
Human milk	Preferred over formula	I/C-LD	Preferred for EN initiation	Recommended/3
Chylothorax	DHM or MCT-rich formula recommended	IIb/B-NR	NS	-
Calories	>90–120 kcal/kg/day (likely higher than in healthy full-term controls). Unpredictable in the perioperative phase and likely to normalize to age-matched controls 1 week postoperatively	IIb/C-EO III/C-LD	EN 120–150 kcal/kg/day	-
Fluids	NS	-	EN 120–140 mL/kg/day	-
Postoperative safety	NS	-	Safe for use in hemodynamically stable patients and should be initiated as soon as possible. Close monitoring is required for patients with GI and cardiac complications.	Strongly recommend/2
Weight monitoring in interstage period	NS	-	Weight monitoring is critical for the early identification of growth faltering and intervention.	Strongly recommended/4
Involvement of other specialists	NS	-	Dietician should be involved at each clinic visit and when nutritional concerns arise.	Strongly recommended/4
Parenteral Nutrition				
Time of the introduction	When EN insufficient	IIb/C-LD	Early introduction	Strongly recommended/2
Delayed strategy	May be beneficial if energy adequacy will be achieved during the first week of admission	IIb/B-R	NS	-
Lipid emulsions	Soy-based lipid emulsion can be used in patients without comorbidities	IIa, C-EO	Start at 1–3 g/kg/day	-
Calories	>90–120 kcal/kg/day	IIb/C-EO	90–100 kcal/kg/day	Recommended/5
Proteins	>2 g/kg/day Unpredictable in the perioperative phase and likely to normalize to age-matched controls 1 week postoperatively	IIb/C-EO III/C-LD	3–4 g/kg/day	Recommended/5
Fluids	NS	-	100–150 mL/kg/day	Recommended/5
Fats	NS	-	3 g/kg/day	Recommended/5
Glucose	NS	-	12–14 g/kg/min	Recommended/5
Vitamins	NS	-	Levo-carnitine: 8–10 mg/kg/day	Recommended/5
Micronutrients (kg/day)	NS	-	Na: 2–5 mEq; K: 2–4 mEq; Ca: 0.5–4 mEq; P: 0.5–2 mMol; Mg: 0.3–0.5 mEq; Zn: 50–250 mcg; Co: 20 mcg; Mn 1 mcg; Se: 2 mcg; Cr: 10–15 mcg	Recommended/5
Weight gain goal	NS	-	20–30 g/day	-

CHD—Congenital heart disease, DHM—defatted human milk, ECMO—extracorporeal membrane oxygenation, EN—enteral nutrition, GI—gastrointestinal, IC—indirect calorimetry, IL—intravenous lipid solution, LOE—level of evidence, MCT—medium-chain triglyceride, NS—not stated, PGE—prostaglandins, PN—parenteral nutrition, REE—resting energy expenditure, TPN—total parenteral nutrition, VAD—ventricular assist device.

## Data Availability

The data presented in this study are available on request from the corresponding author.

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
