# Peer review of "A Systematic Review of Clinical Practice Guidelines on the Management of Malnutrition in Children with Congenital Heart Disease"

_nutrients, 2024, doi:10.3390/nu16162778_

Round 1

Reviewer 1 Report

Comments and Suggestions for Authors

The aim of this paper is to systematically evaluate clinical guidelines for the management of malnutrition in children with CHD, to identify the consistency and quality of existing recommendations, and to promote the development of more comprehensive and practical guidelines to improve clinical outcomes in children. But authors need to consider the following questions.

1. Selection of articles: First of all, literature search ends in September 2023, and article presentation is 2024. Authors should consider whether new relevant studies or guidelines have been published during this period. Second, although multiple databases and websites were searched, only two guidelines were included in the study, suggesting further optimization of search strategies and explicit discussion of whether the guidelines were applicable to different regions. Finally, while the AGREE II tool is used to assess guideline quality, the level and quality of evidence behind each recommendation needs to be clearly communicated.

2. Specificity of the implementation strategy: It was mentioned that the guidance included lacked implementation recommendations or tools, which could be a major deficiency. If there is no clear implementation strategy, it is difficult to apply in clinical practice.

3. Long-term follow-up data: The documentation does not provide sufficient information or data on the long-term effects and impacts of nutritional interventions.

Comments on the Quality of English Language

good

Author Response

Reviewer: 1

The aim of this paper is to systematically evaluate clinical guidelines for the management of malnutrition in children with CHD, to identify the consistency and quality of existing recommendations, and to promote the development of more comprehensive and practical guidelines to improve clinical outcomes in children. But authors need to consider the following questions.

1.Selection of articles: First of all, literature search ends in September 2023, and article presentation is 2024. Authors should consider whether new relevant studies or guidelines have been published during this period. Second, although multiple databases and websites were searched, only two guidelines were included in the study, suggesting further optimization of search strategies and explicit discussion of whether the guidelines were applicable to different regions. Finally, while the AGREE II tool is used to assess guideline quality, the level and quality of evidence behind each recommendation needs to be clearly communicated.

RESPONSE: We updated our search, but none of the new guidelines were identified.

The main reason of such low number of included studies was because one of our inclusion criteria was that it had to include a system for assessing the evidence to create the recommendation. We are aware that drawing conclusions based on a small number of studies is burdened with the risk of bias, but on the other hand, it shows that there is a great need for this type of publication.

In the process of creating guidelines, the authors, performing a systematic review of the literature that will serve them to create recommendations. When creating a review of guidelines, we were focusing exclusively on documents that were guidelines.

The paragraph 3.3 Summary of individual domains – contains summary of each domain of the AGREE II. This tool is focusing mainly on the methodological quality of guidelines not on the level of evidence for each recommendation. In Table 3 we presented the level of evidence and grading for each of the recommendations from the included guidelines.

  1. Specificity of the implementation strategy: It was mentioned that the guidance included lacked implementation recommendations or tools, which could be a major deficiency. If there is no clear implementation strategy, it is difficult to apply in clinical practice.

RESPONSE: We agree with the reviewer. This is one of the most important elements of the guidelines. In the guidelines that were included in our review, this was the most poorly rated domain (Applicability (Domain 5) – 25 and 17% only). That is why it is also one of the main conclusions from our review. To strengthen it, we added the following sentence “Since this is one of the most important components of the guidelines, the new documents should particularly focus on creating an appropriate strategy for implementing the recommendations.”

  1. Long-term follow-up data: The documentation does not provide sufficient information or data on the long-term effects and impacts of nutritional interventions.

RESPONSE: The aim of our review was to check the number of guidelines on the nutrition of children with heart defects and to check their consistency. It was not our aim to assess the long-term impact of the nutritional recommendations.

Reviewer 2 Report

Comments and Suggestions for Authors

In the manuscript submitted to me for review entitled "A systematic review of clinical practice guidelines on the management of malnutrition in children with congenital heart diseasethe authors Maciej KoÅ‚odziej and Julia Skulimowska present a study evaluating recommendations and guidelines for nutritional intervention in children with congenital heart diseases (CHD).

The research compared two studies conducted within a decade of each other, and thus indicated changes and advances in nutritional interventions in newborns with CHD. The information presented is consolidated into 3 tables in the main text of the manuscript, 3 supplementary tables and 1 supplementary figure included in the supplementary file.

The authors have supported their research with 20 references, which mainly present accumulated information from the last decade. About 1/2 of the total number of references are from recent years, indicating that the topic of nutrition in children with CHD has been actively researched by various authors in recent years, and it is likely that this manuscript would be of interest to Nutrients readers.

My remarks and recommendations to the authors are:

1. The authors claim to have searched large databases, but have included only 20 references to support their study. The Introduction section should be expanded a bit and include more references. However, this is a review article, and the Introduction section in this version is almost entirely within the volume of the title page of the manuscript.

2. In the Materials and Methods section, authors refer to themselves as reviewers in the third person plural, as if it were someone else. Somewhat atypical and confusing is such a way of their own participation in writing the manuscript.

3. The authors focus on 2 studies from 2013 and 2022, but give the information as a series of % ratings, which would make it difficult for readers to understand the meaning of the information. If it is possible to present the results a little more descriptively (it is partly done in the discussion, but not enough). Some little information about the two studies is given in Tables 1 and 3. It would be good if this information was given more extensively in the text, for to make clearer to the reader the scientific contribution in the two studies on which this manuscript is based.

4. In the References section, a large part of the references have the first author indicated and follows et al. According to the instructions for authors, this is permissible if there are more than 10 authors, and after the tenth one can indicate et al. The manuscript cannot be based on 2 other articles and not have all authors listed in them. Such are also references with numbers 3, 4, 7, 11, 14, 15, 16 and 17. Let all authors be added (see instructions for authors).

Author Response

Reviewer: 2

In the manuscript submitted to me for review entitled "A systematic review of clinical practice guidelines on the management of malnutrition in children with congenital heart disease“ the authors Maciej KoÅ‚odziej and Julia Skulimowska present a study evaluating recommendations and guidelines for nutritional intervention in children with congenital heart diseases (CHD).

The research compared two studies conducted within a decade of each other, and thus indicated changes and advances in nutritional interventions in newborns with CHD. The information presented is consolidated into 3 tables in the main text of the manuscript, 3 supplementary tables and 1 supplementary figure included in the supplementary file.

The authors have supported their research with 20 references, which mainly present accumulated information from the last decade. About 1/2 of the total number of references are from recent years, indicating that the topic of nutrition in children with CHD has been actively researched by various authors in recent years, and it is likely that this manuscript would be of interest to Nutrients readers.

My remarks and recommendations to the authors are:

  1. The authors claim to have searched large databases, but have included only 20 references to support their study. The Introduction section should be expanded a bit and include more references. However, this is a review article, and the Introduction section in this version is almost entirely within the volume of the title page of the manuscript.

RESPONSE: We added following senteces to the introduction section:

“Despite the fact, that a lot of clinical studies and narrative reviews are being conducted nowadays, there are very few universal, evidence-based guidelines, which could be safely introduced in hospitals worldwide. Additionally, it seems that the least-explored topic is the nutrition of children in the preoperative period.[11] Considering the emerging evidence that proper nutritional preparation of the patient can significantly improve surgical outcomes and shorten the length of hospitalization (which translates to a reduced risk of perioperative and postoperative complications), it appears warranted for new research and guidelines to address the preoperative period in more detail. [12]

  1. In the Materials and Methods section, authors refer to themselves as reviewers in the third person plural, as if it were someone else. Somewhat atypical and confusing is such a way of their own participation in writing the manuscript.

RESPONSE: We are sorry for the confusion however, we used standard phrases which are mostly required by the PRISMA statement.

  1. The authors focus on 2 studies from 2013 and 2022, but give the information as a series of % ratings, which would make it difficult for readers to understand the meaning of the information. If it is possible to present the results a little more descriptively (it is partly done in the discussion, but not enough). Some little information about the two studies is given in Tables 1 and 3. It would be good if this information was given more extensively in the text, for to make clearer to the reader the scientific contribution in the two studies on which this manuscript is based.

RESPONSE:  Please find corrected paragraph 3.3 which was added to the manuscript:

3.3 Summary of individual domains

Scope and purpose (Domain 1)

In this domain, both guidelines achieved the highest score of 100%. In both guidelines the overall objectives and health questions covered by the guidelines as well as the target population were specifically described.

Stakeholder involvement (Domain 2)

In the stakeholder domain, Mills et al. [14] received a slightly higher score of 37% compared to 31% obtained by Slicker et al. [13] The reason for such results was a lack of the views and preferences of the target population and a lack of a clear definition of the guideline target users, where both Mills et al. and Slicker et al. revceived 2 out of 7 points in all assessments. There was a 1 point difference in the scoring of Item 4, i.e. the inclusion of individuals from all relevant professional groups in the process of the guideline development, where Mills et al. received 5 and Slicker et al. 4 out of maximum 7 points.

Rigor of development (Domain 3)

Both guidelines received comparable scores, for Slicker et al. [13] of 67% and for Mills et al. [14] of 72%. The main difference was in Item 9, which describes the strengths and limitations of the body of evidence, where Mills et al. received on average 2 points more than Slicker et al. Also in Item 8, which assesses the quality of the criteria for selecting evidence, Millis et al. received maximum score comparing to Slicker et al., who got one point less. Otherwise in Items 7 (use of systematic methods to search for evidence), 10 (description of methods for formulating the recommendations), 11 (consideration of the health benefits, side effects and risks) and 12 (linkage between the recommendations and the supporting evidence) both Millis et al. and Slicker et al. received the maximum of 7 points. In both Items 13 and 14 both Millis et al. and Slicker et al. received 1 point as the guidelines were not reviewed externally prior to their publication, and no information about updating the guidelines was included.

Clarity of presentation (Domain 4)

Both guidelines were noted high in the clarity of presentation domain (94% and 100%). The difference was observed by only 1 point in the assessment of Item 15. That indicated a slightly higher specificity and ambiguity of guidelines presented by Mills et al. [14].

Applicability (Domain 5)

Mills et al. [14] received a slightly higher score in this domain of 25% compared to 17% obtained by Slicker et al. [13]. However, in both guidelines, this domain was the one, which received the lowest score. This was due to the fact that none of the guidelines provided neither advice and/or tools on how the recommendations can be put into practice (Item 19), potential resource implications of applying the recommendations (Item 20), nor the monitoring and/or auditing criteria (Item 21). The little difference in scoring lied in the assessment of Item 18, assessing facilitators and barriers to the application of the guideline, where Millis et al. received 7/7 points and Slicker et al. 5 points.

Editorial independence (Domain 6)

In this domain, both guidelines achieved the highest score of 100%. In both guidelines the views of the funding body had no influence on the content of the guideline (Item 22) and there were no relevant conflicts of interest (Item 23).

Overall quality score

Mills et al. [14] received a higher score in the overall quality (83%) compared to Slicker et [13] al., who received 67%. Slicker et al [13] scored 111 points for the maximum of 161 points, comparing to Mills et al. [14], who received 118 points. For the specific scoring please see Table 2.

  1. In the References section, a large part of the references have the first author indicated and follows et al. According to the instructions for authors, this is permissible if there are more than 10 authors, and after the tenth one can indicate et al. The manuscript cannot be based on 2 other articles and not have all authors listed in them. Such are also references with numbers 3, 4, 7, 11, 14, 15, 16 and 17. Let all authors be added (see instructions for authors).

RESPONSE: Corrected.